# Hiding in Plain Sight: Tweets with Hate Speech Masked by Homoglyphs

**Portia Cooper, Mihai Surdeanu, Eduardo Blanco**
University of Arizona, Tucson, AZ, USA
portiacooper@arizona.edu

## Abstract

To avoid detection by current NLP monitoring applications, progenitors of hate speech often replace one or more letters in offensive words with homoglyphs, visually similar Unicode characters. Harvesting real-world hate speech containing homoglyphs is challenging due to the vast replacement possibilities. We developed a character substitution scraping method and assembled the Offensive Tweets with Homoglyphs (OTH) Dataset[1] (N=90,788) with more than 1.5 million occurrences of 1,281 non-Latin characters (emojis excluded). In an annotated sample (n=700), 40.14% of the tweets were found to contain hate speech. We assessed the performance of seven transformer-based hate speech detection models and found that they performed poorly in a zero-shot setting (F1 scores between 0.04 and 0.52), but normalizing the data dramatically improved detection (F1 scores between 0.59 and 0.71). Training the models using the annotated data further boosted performance (highest micro-averaged F1 score=0.88, using five-fold cross validation). This study indicates that a dataset containing homoglyphs known and unknown to the scraping script can be collected, and that neural models can be trained to recognize camouflaged real-world hate speech.

## 1   Introduction

*Disclaimer: This paper includes language that some readers might find offensive.*

Hate speech, discriminatory language against individuals or groups based on race/ethnicity, nationality, gender, religion, LGBTQ+ identity, or disability status, is banned by Facebook, YouTube, and other major platforms. A common strategy to mask hate speech is replacing one or more letters in offensive words with homoglyphs, Unicode characters that are visually homogeneous (Boucher et al.,

2021). For instance, the Latin "a" (U+0061) and the Cyrillic "а" (U+0430) are nearly indistinguishable to the human eye, yet they belong to different Unicode character families. Currently, there are almost 150,000 Unicode characters,[2] which presents extensive substitution possibilities.

Despite the prevalence of online hate speech containing homoglyphs, the collection of organically generated data of this type is rare. To remedy this scarcity, we developed a dataset of real-world offensive text containing homoglyphs and used this data to train existing hate speech detection models. In particular, our contributions are:

- Developing a novel character substitution scraping method used to assemble a dataset of 90,788 tweets with offensive words containing homoglyphs. To our knowledge, our dataset is the first to be composed of real-world, homoglyph-laden texts.

- Evaluating the effectiveness of replacing Latin characters with homoglyphs as an obfuscation strategy by testing the zero-shot performance of seven open-source transformer-based hate speech detection models hosted by Hugging Face on three versions of an annotated sample of the dataset: (1) original with homoglyphs, (2) partially normalized (Cyrillic homoglyphs replaced with the corresponding Latin letters), and (3) fully normalized (all non-Latin homoglyphs replaced with the corresponding Latin characters).

- Demonstrating that models can be trained to recognize real-world hate speech containing homoglyphs known and unknown to the scraping script.

---

[1] https://github.com/pcoopercoder/
Offensive-Tweets-with-Homoglyphs-OTH-Dataset

[2] https://www.unicode.org/versions/stats/chart_charbyyear.html

## 2 Related Work

Previous work that explored human-generated text with homoglyphs includes the ANTHRO algorithm, which evaluated the phonetic properties of words to locate character-based perturbations, including homoglyphs (Le et al., 2022). Boucher et al. (2021) used homoglyphs, letter reordering, and letter deletion to test the detection capabilities of NLP models. Additionally, Kurita et al. (2019) artificially produced offensive text containing homoglyphs to simulate adversarial attacks.

Woodbridge et al. (2018) investigated obfuscation mitigation using a Siamese convolutional neural network to convert homoglyphs into the characters they resembled. Similarly, Ginsberg and Yu (2018) proposed a "visual scanning" method to detect and predict homoglyphs. Other work has inventoried homoglyphs (Suzuki et al., 2019) and identified previously unknown homoglyphs (Deng et al., 2020).

Finally, several studies have evaluated phishing attacks in which homoglyphs were used to imitate corporate domain names to deceive users and extract personal information (Maneriker et al., 2021; Lee et al., 2020; Wolff and Wolff, 2020). These studies involved training models using artificially created data containing homoglyphs and did not evaluate real-world content.

## 3 Approach

### 3.1 Scraping Twitter

Tweets were collected from Twitter (now renamed X) in December 2022 using the Python Tweepy Library (computational cost detailed in Appendix A.1). Scripting was used to create query terms derived from the 41 most offensive American English words ranked by native English-speaking college students at a large U.S. metropolitan university (Bergen, 2016) (list in Appendix A.2). Query terms were generated by replacing each Latin letter in the offensive words with the corresponding Cyrillic homoglyph(s) from the Unicode Confusable Standard[3] (Table 1). The Latin letters "b", "h", "i", "w", and "y" had multiple Cyrillic homoglyphs, thus words containing these letters generated more than one query term for each letter. For example, the word "bitch" yielded eight variations (ʙitch, Ьitch, bitch, bItch, biтch, bitch, bitcн, and bitch). The

[3] https://www.unicode.org/Public/security/revision-03/confusablesSummary.txt

| Latin Letter | Unicode Value | Cyrillic Homoglyph(s) | Unicode Value(s) |
|---|---|---|---|
| a | U+0061 | а | U+0430 |
| b | U+0062 | в, ь | U+0432, U+044C |
| c | U+0063 | с | U+0441 |
| d | U+0064 | ԁ | U+0501 |
| e | U+0065 | е | U+0435 |
| f | U+0066 | - | - |
| g | U+0067 | Ԍ | U+050D |
| h | U+0068 | н, һ | U+043D, U+04BB |
| i | U+0069 | і, Ӏ | U+0456, U+04CF |
| j | U+006A | ј | U+0458 |
| k | U+006B | к | U+043A |
| l | U+006C | - | - |
| m | U+006D | м | U+043C |
| n | U+006E | п | U+043F |
| o | U+006F | о | U+043E |
| p | U+0070 | р | U+0440 |
| q | U+0071 | ԛ | U+051B |
| r | U+0072 | г | U+0433 |
| s | U+0073 | ѕ | U+0455 |
| t | U+0074 | т | U+0442 |
| u | U+0075 | ц | U+0446 |
| v | U+0076 | ѵ | U+0475 |
| w | U+0077 | ԝ, ѡ | U+051D, U+0461 |
| x | U+0078 | х | U+0445 |
| y | U+0079 | ү, у | U+04AF, U+0443 |
| z | U+007A | - | - |

Table 1: Unicode values of Latin letters and corresponding Cyrillic homoglyphs.

Latin letters "f", "l", and "z" had no Cyrillic homoglyphs thus yielded no variations. For example, the word "fuck" yielded only three query terms (fцck, fucк, and fucк). This process produced 247 query terms that included 26 of the 28 Cyrillic homoglyphs for Latin letters, as "v" and "x" were not present in the 41 offensive words used to generate the query terms.

A total of 93,042 tweets were collected, including 2,254 duplicates created by the presence of multiple query terms in a single tweet. Duplicate tweets were removed, resulting in 90,788 tweets, which we named the Offensive Tweet with Homoglyphs (OTH) Dataset. Figure 1 provides an example of a homoglyph-laden tweet included in the OTH Dataset. The metadata for each tweet was collected from Twitter for aggregate analysis purposes. We also calculated the number of unique non-Latin characters (emojis excluded) present in the dataset and the number of times each was detected.

### 3.2 Annotation

From the OTH Dataset, a random sample of 700 tweets was selected. Using a detailed codebook (Appendix A.3), two human annotators independently evaluated the tweets in the sample (annotator

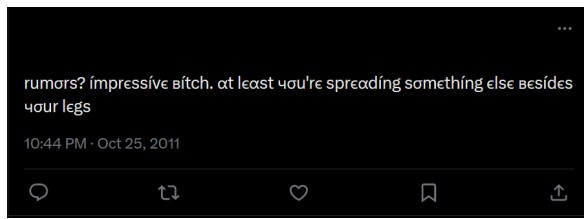

Figure 1: Homoglyph-laden tweet in the Offensive Tweets with Homoglyphs (OTH) Dataset.

information in Appendix A.4), and an IRB exemption for research not involving human participants was granted.

Intercoder agreement exceeded 96.00% for all codes, and Cohen's Kappa ranged from 0.80 to 1.00 (Appendix A.5, Table 5). Coding disagreements were discussed and reconciled.

### 3.3 Tweet Normalization

A survey of existing homoglyph normalization resources was performed, and widely used Python libraries were evaluated (Appendix A.6). No tool was found which automated the conversion of all possible homoglyphs into the Latin characters they resemble. Normalization in the present study was accomplished using scripting and manual compilation. Two normalized versions of the annotated sample were created: (1) partially normalized (Cyrillic homoglyphs replaced with the corresponding Latin letters using custom scripting) and (2) fully normalized (all non-Latin homoglyphs replaced with the corresponding Latin characters using manual compilation).

### 3.4 Base Models

The original, partially normalized, and fully normalized versions of the annotated sample were run separately through seven open-source transformer-based hate speech detection models hosted by Hugging Face (computational cost detailed in Appendix A.7). All selected models were trained (at least in part) on hate speech collected from social media platforms (a summary of each of the utilized models is included in Appendix A.8). Accuracy, precision, recall, and F1-score were calculated.

### 3.5 Five-Fold Cross Validation

Finally, five-fold cross validation was performed on each of the seven models using the original version of the annotated sample to evaluate the extent to which the models learned when exposed to real-world data with homoglyphs (computational cost

| Characters | OTH Dataset N=90,788 | Annotated Sample n=700 |
|---|---|---|
| **Cyrillic homoglyphs for Latin letters** | | |
| Unique characters detected | 27 | 21 |
| Occurrences | 1,264,406 | 9,759 |
| **Additional Cyrillic characters** | | |
| Unique characters detected | 72 | 35 |
| Occurrences | 125,973 | 1,129 |
| **Other non-Latin characters*** | | |
| Unique characters detected | 1,182 | 93 |
| Occurrences | 136,346 | 1,164 |
| **Total** | | |
| Unique characters detected | 1,281 | 149 |
| Occurrences | 1,526,725 | 12,052 |

*Emojis excluded

Table 2: Character composition of the Offensive Tweets with Homoglyphs (OTH) Dataset and the annotated sample.

detailed in Appendix A.7). The micro-averages of accuracy, precision, recall, and F1-score across the five folds were calculated.

## 4 Results

### 4.1 OTH Dataset

The 90,788 tweets in the OTH Dataset were posted by 31,878 unique author IDs. The dataset included 1,264,406 occurrences of 27 Cyrillic homoglyphs for Latin letters [only the Cyrillic "q" (U+051B) was not detected] (Table 2). Importantly, 72 *additional* Cyrillic characters occurred 125,973 times, and 1,182 other non-Latin characters (emojis excluded) occurred 136,346 times. The dataset included an average of 16.82 Cyrillic and other non-Latin characters per tweet.

As shown in Figure 2, the homoglyphs with the highest number of occurrences were the Cyrillic "e" (n=295,039, U+0435), "o" (n=238,643, U+043E), and "a" (n=225,668, U+0430). The non-Cyrillic homoglyphs with the highest number of occurrences were "í" (n=26,246, U+00ED), "$\sigma$" (n=24,730, U+03C3), and "$\alpha$" (n=23,465, U+03B1).

Of the 247 query terms searched, 156 returned tweets. The earliest tweet in the OTH Dataset was posted on February 19, 2009 (31 months after the inception of Twitter), and the most recent tweet was posted on December 20, 2022. The tweets were assigned 43 language identifiers by Twitter, and the majority were classified as Czech (62.04%). Only 24.27% were classified as English by Twitter (Appendix A.9, Table 6), even though all tweets

| Model | Original Version | | | | Partially Normalized Version | | | | Fully Normalized Version | | | |
|---|---|---|---|---|---|---|---|---|---|---|---|---|
| | A | P | R | F1 | A | P | R | F1 | A | P | R | F1 |
| 1 | 0.60 | 0.67 | 0.02 | 0.04 | 0.71 | **0.79** | 0.38 | 0.52 | 0.74 | **0.81** | 0.48 | 0.60 |
| 2 | 0.60 | 0.49 | 0.16 | 0.24 | 0.65 | 0.55 | 0.65 | 0.60 | 0.69 | 0.58 | 0.78 | 0.67 |
| 3 | **0.65** | **0.83** | 0.17 | 0.29 | **0.75** | **0.79** | 0.52 | 0.63 | **0.78** | 0.77 | 0.63 | 0.70 |
| 4 | 0.56 | 0.46 | 0.46 | 0.46 | 0.50 | 0.44 | **0.91** | 0.60 | 0.53 | 0.46 | **0.98** | 0.62 |
| 5 | 0.58 | 0.47 | 0.40 | 0.43 | 0.60 | 0.50 | 0.73 | 0.59 | 0.63 | 0.54 | 0.66 | 0.59 |
| 6 | 0.57 | 0.47 | **0.57** | **0.52** | 0.66 | 0.55 | 0.84 | **0.67** | 0.70 | 0.59 | 0.83 | 0.69 |
| 7 | 0.63 | 0.54 | 0.44 | 0.49 | 0.66 | 0.55 | 0.80 | 0.65 | 0.72 | 0.61 | 0.84 | **0.71** |

Table 3: Accuracy (A), precision (P), recall (R), and F1-scores (F1) for seven hate speech detection models on three versions of the annotated sample (n=700) of the Offensive Tweets with Homoglyphs (OTH) Dataset: (1) original with homoglyphs, (2) partially normalized (Cyrillic homoglyphs replaced with the corresponding Latin letters), and (3) fully normalized (all non-Latin homoglyphs replaced with the corresponding Latin characters).

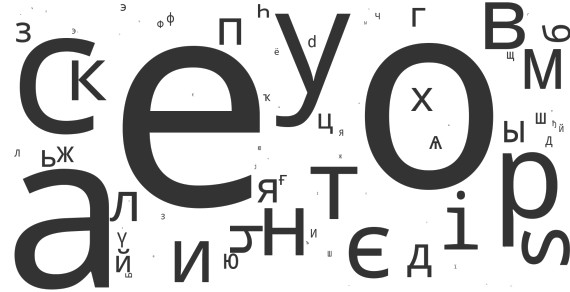

Figure 2: Word cloud representation of 99 Cyrillic characters present in the Offensive Tweets with Homoglyphs (OTH) Dataset by occurence (n=1,390,379).

in the OTH Dataset included at least one English-language query term.

## 4.2 Annotated Sample

The annotated sample resembled the OTH Dataset in terms of character composition and included an average of 17.22 Cyrillic and other non-Latin characters per tweet.

In the annotated sample, 40.14% of tweets were classified as hate speech by human annotators. Most of these tweets (87.54%) included misogynistic hate speech. Tweets referencing hate speech related to LGBTQ+ identity (5.34%), race/ethnicity (3.91%), disability status (1.42%), and religion (1.07%) were less common. No hate speech related to nationality was found in the sample. Additionally, 97.71% of tweets were classified as offensive; 20.43% were labeled sexually explicit; and 5.14% referenced violence or aggressive acts. The body text of the vast majority of tweets (92.43%) was classified as English by the annotators.

## 4.3 Zero-Shot Model Performance

On the original version of the annotated sample with homoglyphs, the F1-scores of the seven hate speech detection models ranged from 0.04 to 0.52 (Table 3). On the partially normalized version of the annotated sample (Cyrillic homoglyphs replaced with the corresponding Latin letters), F1-scores ranged from 0.52 to 0.67. On the fully normalized version of the annotated sample (all non-Latin characters replaced with the corresponding Latin characters), F1-scores ranged from 0.59 to 0.71.

## 4.4 Five-Fold Cross Validation

In the five-fold cross validation on the annotated sample with homoglyphs, F1-scores ranged from 0.41 to 0.88 (Table 4).

| Model | A | P | R | F1 |
|---|---|---|---|---|
| 1 | 0.90 | 0.88 | 0.86 | 0.87 |
| 2 | 0.69 | 0.84 | 0.27 | 0.41 |
| 3 | 0.81 | 0.77 | 0.76 | 0.77 |
| 4 | **0.91** | **0.91** | 0.86 | **0.88** |
| 5 | 0.90 | 0.88 | 0.86 | 0.87 |
| 6 | 0.89 | 0.87 | 0.85 | 0.86 |
| 7 | 0.89 | 0.85 | **0.88** | 0.87 |

Table 4: Accuracy (A), precision (P), recall (R), and F1-scores (F1) across five-fold cross validation for seven hate speech detection models on the annotated sample (n=700) of the Offensive Tweets with Homoglyphs (OTH) Dataset.

## 5 Discussion

Our character substitution scraping method yielded 90,788 tweets containing more than 1.5 million occurrences of 1,281 non-Latin characters (emojis

excluded). The search strategy used Cyrillic homoglyphs for Latin letters to assemble a broader collection of non-Latin characters. As expected, the bulk of the OTH Dataset (82.82%) was comprised of Cyrillic homoglyphs. However, 262,319 occurrences of 1,254 other non-Latin characters were also captured, including 72 additional Cyrillic characters and 1,182 characters from other Unicode character families. The most common Cyrillic characters were "а" (U+0430), "е" (U+0435), and "о" (U+043E), which are homoglyphs for the Latin letters "a", "e", and "o." The most common non-Cyrillic characters were "$\alpha$" (U+03B1), "í" (U+00ED), and "$\sigma$" (U+03C3), which are homoglyphs for the Latin letters "a", "i", and "o". These results may reflect malicious users' preference for homoglyphs that mimic Latin vowels.

The noise produced by the homoglyphs impeded the performance of all seven hate speech detection models tested. Fully normalizing the data by replacing all non-Latin characters with the corresponding Latin characters dramatically improved model performance. Most notably, Model 1's F1-score jumped from 0.04 to 0.60.

In the five-fold cross validation, five models achieved F1-scores that exceeded 0.85. Conversely, Model 2 performed poorly (F1-score = 0.41). This may be related to the model's original training data, which included only hate speech labeled as LGBTQ+ identity and nationality–categories of hate speech which were rare within the annotated sample in the present study. The performance of the other six models demonstrates that neural classifiers can be trained to recognize real-world hate speech masked by homoglyphs. This is an exciting result, considering that the dataset contains many homoglyphs unknown to the scraping script, and thus these perturbations could not be addressed through deterministic normalization.

The annotated sample analyzed in the present study included a 40%-60% split between tweets with and without hate speech. The large volume of hate speech tweets with misogynistic content (87.54%) found in the annotated sample is notable. This result is consistent with a prior survey of Twitter that found 419,000 female slurs were posted on average each day (Felmlee et al., 2020).

Homoglyphs also appeared to interfere with Twitter's internal system that classifies the body text language of tweets. In the annotated sample, Twitter assigned the English language identifier

to only 21.00% of tweets, but human annotators found that 92.43% of the tweets were written in English. Twitter classified the majority of tweets in the annotated sample (60.50%) and the OTH Dataset (62.04%) as Czech. These results are especially interesting because the Czech language does not include Cyrillic characters, which were used to generate the query terms.

# 6 Conclusion

Infusing hate speech with homoglyphs was found to be an effective strategy to avoid detection. In the present study, existing hate speech detection models were ineffective against obfuscation of this type. The OTH Dataset offers training data to build hate speech detection models inoculated against real-world homoglyphs.

# 7 Limitations

Cyrillic homoglyphs for Latin letters were used to generate the query terms in the present study because they are widely used and were exempt from the normalization that Twitter performs on query terms prior to searching its repository. A more diverse dataset may be achieved by expanding the search strategy to include (1) homoglyphs from multiple Unicode character families, (2) complex homoglyph substitution combinations in query terms, and (3) a broader list of offensive words to generate query terms.

# 8 Ethical Consideration

Due to the offensive nature of the OTH Dataset and the data sharing policy of X (Twitter), only the Tweet IDs of the OTH Dataset will be published.

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

# A Appendix

## A.1 Computational Costs of Scraping Twitter

An estimated 48 total hours were necessary to compile the OTH Dataset from the Twitter corpus. Tweet scraping was performed locally.

## A.2 41 Most Offensive American English Words Reported by Bergen (2016)

asshole, bastard, bitch, blowjob, buttfuck, chink, clit, cock, cocksucker, cunt, dick, dumb, dyke, fag, fuck, gay, goddamn, gook, homo, hooker, kike, lesbo, loser, moron, motherfucker, nigger, nutsack, prick, pussy, queer, retard, rimjob, shit, shithead, skank, slut, sodomize, spic, tits, twat, whore

## A.3 Codebook for Annotating Sample of the OTH Dataset

1. English Lanuage Body Text
   Includes predominately English language body text.
   0=No
   1=Yes

2. Sexually Explicit Content
   Contains graphic sexual description of genitalia or sex acts, such as intercourse, oral sex, and masturbation.
   0=No
   1=Yes

3. References to Violence or Aggressive Acts
   References physical violence or aggressive behavior.
   0=No
   1=Yes

4. Offensive Content
   Contains profanity or rude language. This category includes hate speech, as well as any other content that is offensive.
   0=No
   1=Yes

5. Hate Speech
Contains rhetoric that is derogatory or promotes, rationalizes, or reinforces hatred towards a target group or individual based on protected characteristics. Protected characteristics include gender, race/ethnicity, LGBTQ+ identity, nationality, disability status, and religion. Slurs that are used to attack people based on their protected characteristics should be classified as hate speech.
0=No
1=Yes

6. Category of Hate Speech (Select All That Apply)
*Complete only for tweets classified as hate speech (Codebook Section 5).*

   (a) Gender - hate speech pertaining to a group of people classified under a specific gender identity.
   0=No
   1=Yes

      *Complete only for tweets classified as gender hate speech*
      i. Misogyny - hate speech that exhibits hatred of, contempt for, or prejudice against women
      0=No
      1=Yes

   (b) Race/Ethnicity - hate speech pertaining to a group of people who possess a shared cultural background.
   0=No
   1=Yes

   (c) LGBTQ+ identity - hate speech directed at the LGBTQ+ community.
   0=No
   1=Yes

   (d) Nationality - hate speech pertaining to people from a specific nation.
   0=No
   1=Yes

   (e) Disability Status - hate speech pertaining to people who have physical or mental conditions that limits their movements, senses, or activities.
   0=No
   1=Yes

   (f) Religion - hate speech pertaining to the worship of God or other superhuman entities.

0=No
1=Yes

## A.4 Annotators for Random Sample of the OTH Dataset

The lead investigator and an experienced annotator coded the annotated sample of the OTH Dataset. No identifying information from the annotators was collected.

## A.5 Intercoder Agreement

| Code | Intercoder Agreement (%) | Cohen's Kappa | SE | 95% CI |
|---|---|---|---|---|
| Hate speech | 97.14 | 0.94 | 0.01 | 0.91 – 0.97 |
| Gender | 98.22 | 0.88 | 0.05 | 0.78 – 0.98 |
| LGBTQ+ | 98.21 | 0.81 | 0.08 | 0.64 – 0.97 |
| Race | 98.93 | 0.84 | 0.09 | 0.66 – 1.00 |
| Disability* | 99.64 | 0.89 | 0.12 | 0.67 – 1.00 |
| Religion | 99.64 | 0.80 | 0.20 | 0.41 – 1.00 |
| Nationality* | - | - | - | - |
| Offensive content | 99.56 | 0.91 | 0.05 | 0.80 – 1.00 |
| Sexually explicit | 96.44 | 0.89 | 0.02 | 0.84 – 0.93 |
| References violence or aggressive acts | 98.43 | 0.82 | 0.05 | 0.71 – 0.92 |
| English-language body text | 100.00 | 1.00 | 0.00 | 1.00 – 1.00 |

*No tweets containing hate speech related to nationality were identified in the annotated sample.

Table 5: Intercoder agreement by annotation code in the annotated sample (n=700) of the Offensive Tweets with Homoglyphs (OTH) Dataset.

## A.6 Evaluation of Existing Normalization Tools

1. `Unidecode` library[4] focuses on transliteration conversions rather than homoglyph normalization. This introduces several character edge cases. For example, the Cyrillic "п" (U+043F), which is classified as a homoglyph for the Latin "n" (U+006E) by the Unicode Confusable standard, is normalized by Unidecode to the Latin letter "p" (U+0070).

2. `cyrtranslit` library[5] only performs bidirectional Cyrillic to Latin text and vice versa. As shown in Table 2, the OTH Dataset contains 1,182 unique non-Cyrillic Unicode characters.

3. `confusable_homoglyphs` library[6] focuses on confusable detection as opposed to homoglyph normalization. It offers no normalization function/utility.

---

[4] https://pypi.org/project/Unidecode/
[5] https://pypi.org/project/cyrtranslit/
[6] https://pypi.org/project/confusable_homoglyphs/

4. `confusables` library[7] is an expanded version of the `confusable_homoglyphs` library, and it includes a normalization function. But it struggled to normalize select characters such as the Cyrillic "п" (U+043F), a homoglyph for the Latin "n" (U+006E).

5. `homoglyphs` library[8] includes homoglyph normalization capabilities via a `to_ascii()` function. Unfortunately, `to_ascii()` deletes any characters "which can't be converted by default." This resulted in the deletion of most Cyrillic homoglyphs.

## A.7 Computational Costs of Evaluating and Fine-Tuning Models

Due to the small size of the annotated sample, an estimated maximum four total hours of GPU usage were necessary for this study. Models were run using Cloud-based GPU resources.

## A.8 Selected Open-Source Transformer-Based Hate Speech Detection Models

1. RoBERTa-base binary classification model trained on 58 million tweets (Barbieri et al., 2020)

2. RoBERTa-base binary classification model by Liu et al. (2019) trained on the English subset of the FRENK Dataset (Ljubešić et al., 2019)

3. BERT-base binary classification model trained on data from Twitter and Stormfront, a popular white supremacist forum (Aluru et al., 2020)

4. RoBERTa-base binary classification model trained on 11 English hate speech datasets (Vidgen et al., 2021)

5. RoBERTa-base binary classification model trained on 11 English hate speech datasets and Round 1 of the Dynamically Generated Hate Speech Dataset (Vidgen et al., 2021)

6. RoBERTa-base binary classification model trained on 11 English hate speech datasets and Rounds 1 and 2 of the Dynamically Generated Hate Speech Dataset (Vidgen et al., 2021)

7. RoBERTa-base binary classification model trained on 11 English hate speech datasets

and Rounds 1, 2, and 3 of the Dynamically Generated Hate Speech Dataset (Vidgen et al., 2021)

## A.9 Language Identifiers Assigned by Twitter to Dataset

| Language Identifier | Language | OTH Dataset N=90,788 | Annotated Sample n=700 |
|---|---|---|---|
| cs | Czech | 62.04% | 60.50% |
| en | English | 24.27% | 21.00% |
| und | Undetermined | 5.20% | 0.00% |
| ru | Russian | 5.09% | 4.00% |
| vi | Vietnamese | 4.10% | 6.50% |
| et | Estonian | 0.31% | 0.00% |
| pt | Portuguese | 0.21% | 0.00% |
| uk | Ukrainian | 0.18% | 0.50% |
| tr | Turkish | 0.13% | 1.00% |
| ar | Arabic | 0.13% | 0.50% |
| eu | Basque | 0.09% | 0.00% |
| ca | Catalan | 0.09% | 0.50% |
| tl | Tagalog | 0.08% | 0.00% |
| es | Spanish | 0.08% | 0.00% |
| in | Indonesian | 0.08% | 0.00% |
| fr | French | 0.06% | 0.00% |
| bg | Bulgarian | 0.04% | 0.00% |
| ro | Romanian | 0.04% | 0.00% |
| nl | Dutch | 0.04% | 0.00% |
| de | German | 0.03% | 0.00% |
| sr | Serbian | 0.02% | 0.00% |
| fi | Finnish | 0.02% | 0.00% |
| is | Icelandic | 0.02% | 0.00% |
| pl | Polish | 0.02% | 0.50% |
| qst | Very short text | 0.01% | 0.00% |
| zxx | No linguistic content | 0.01% | 0.00% |
| qht | Hashtag only | 0.01% | 0.00% |
| hu | Hungarian | 0.01% | 0.50% |
| zh | Chinese | 0.01% | 0.00% |
| it | Italian | 0.01% | 0.00% |
| da | Danish | 0.01% | 0.00% |
| th | Thai | 0.01% | 0.00% |
| sv | Swedish | 0.01% | 0.00% |
| ht | Haitian | 0.01% | 0.00% |
| lv | Latvian | 0.01% | 0.00% |
| qme | Media link | 0.01% | 0.00% |
| sl | Slovenian | 0.01% | 0.00% |
| no | Norwegian | 0.00% | 0.00% |
| cy | Welsh | 0.00% | 0.00% |
| lt | Lithuanian | 0.00% | 0.00% |
| ur | Urdu | 0.00% | 0.00% |
| hi | Hindi | 0.00% | 0.00% |
| hy | Armenian | 0.00% | 0.00% |

Table 6: Language identifiers assigned by Twitter to tweets in the Offensive Tweets with Homoglyphs (OTH) Dataset and the annotated sample.

---

[7] https://pypi.org/project/confusables/
[8] https://pypi.org/project/homoglyphs/