# OpenReview forum: "Hiding in Plain Sight: Tweets with Hate Speech Masked by Homoglyphs"
_EMNLP/2023/Conference — EMNLP 2023 Findings_

### Official Review · Reviewer_BrA1 · 2023-08-04

**Soundness:** 3

**Excitement:**

3: Ambivalent: It has merits (e.g., it reports state-of-the-art results, the idea is nice), but there are key weaknesses (e.g., it describes incremental work), and it can significantly benefit from another round of revision. However, I won't object to accepting it if my co-reviewers champion it.

**Paper Topic And Main Contributions:**

This study introduces Offensive Tweets with Homoglyphs (OTH), a pioneering benchmark dataset for Hate Speech Detection concealed by Homoglyphs in code-mixed language. The authors evaluated the performance of seven transformer-based hate speech detection models and observed a significant improvement in detection by normalizing the data (F1 scores ranging from 0.64 to 0.74). These findings demonstrate the feasibility of collecting a dataset comprising both known and unknown homoglyphs, and the efficacy of training neural models to identify camouflaged real-world hate speech.

**Questions For The Authors:**

- Why the author decide to collect data from  create query terms derived from the 41 most offensive American English words.
- The authors state in the Section 3.1 “We also calculated the number of unique non-Latin characters (emojis excluded) present in the dataset and the number of times they were detected”. Please clarify why this sentence is stated here and where the calculated results is presented.
- In section 3.2, author state that "no library currently exists which automates the conversion of every homoglyph into the Latin characters they resemble". However, I found the publicy project that provide a big list of homoglyphs and code to detect them (https://github.com/codebox/homoglyph). So, what is the different between the contribution of authors with this project.
- The authors should provide detailed information regarding the process and properties of the three dataset configurations used for experiments.
- In order to demonstrate that the dataset is useful, comparison to some popular datasets is necessary to highlight the different dimensions that the corpus can contribute to which kind of research.
- It would be valuable for the authors to explain the decision-making process behind the use of the baseline models. This would help researchers to understand the benefits and limitations of these models and potentially apply them to their own research.
- The authors present the performance of models in Table 2, 3, which is significant. Likewise, it would be interesting to explore similar results comparing monolingual and multilingual models.
- Please include annotation guidelines that could help other researchers develop similar guidelines for Hate Speech Masked by Homoglyphs in other languages/domains, etc.

**Reasons To Accept:**

- The paper introduces the Offensive Tweets with Homoglyphs (OTH) dataset, which has the potential to serve as a novel benchmark for Hate Speech Detection obscured by Homoglyphs.

- In addition to providing this dataset, the paper presents a series of compelling experiments with thorough analyses. The authors also offer valuable insights into potential directions for future research in their analysis.

**Reasons To Reject:**

- Why the author decide to collect data from  create query terms derived from the 41 most offensive American English words.
- The statement in Section 3.1 (line 138) regarding the calculation of the number of unique non-Latin characters (excluding emojis) in the dataset, and their frequency of occurrence, should be clarified, along with the presentation of the calculated results.
- In Section 3.2, the authors claim that no existing library automates the conversion of homoglyphs into Latin characters they resemble. However, a publicly available project (https://github.com/codebox/homoglyph) provides a significant list of homoglyphs and code for detection. The authors should explain the differences between their contribution and this existing project.
- Detailed information about the process and properties of the three dataset configurations used for experiments should be provided to ensure transparency and reproducibility.
- In order to demonstrate that the dataset is useful, comparison to some popular datasets is necessary to highlight the different dimensions that the corpus can contribute to which kind of research.
- It would be valuable for the authors to explain the decision-making process behind the use of the baseline models. This would help researchers to understand the benefits and limitations of these models and potentially apply them to their own research.
- The authors present the performance of models in Table 2, 3, which is significant. Likewise, it would be interesting to explore similar results comparing monolingual and multilingual models.
- Please include annotation guidelines that could help other researchers develop similar guidelines for Hate Speech Masked by Homoglyphs in other languages/domains, etc.

**Reproducibility:**

3: Could reproduce the results with some difficulty. The settings of parameters are underspecified or subjectively determined; the training/evaluation data are not widely available.

**Reviewer Confidence:**

5: Positive that my evaluation is correct. I read the paper very carefully and I am very familiar with related work.

**Typos Grammar Style And Presentation Improvements:**

- Section 3.3 could be placed in appendix. Instead, the authors should include statistics on the dataset's labels, distributions, and characteristics.
- To make the analysis of the results more transparent, authors should present their results in several metrics and as a 4-digit decimal (ex: 0.81xx).
- Please Release the dataset into the public domain.
- Please Release the code in the public domain.

---

> ### Author Rebuttal · Authors · 2023-08-28
>
> Thank you for the detailed review of the paper. We are happy to answer your questions:
>
> A. The Bergen list of 41 offensive words (2016) was chosen as it provides empirically established offensive terms. While other lists of offensive terms are available (e.g. https://github.com/LDNOOBW/List-of-Dirty-Naughty-Obscene-and-Otherwise-Bad-Words), we found no other list that had been tested with human respondents.
>
> B. The quote from Section 3.1 was provided to explain our approach (methods). Then, we present these data in the Results (first paragraph of Section 4.1). We are happy to remove this sentence from Section 3.1.
>
> C. While our work discusses homoglyph normalization, the key contribution of our study is the creation of a novel dataset containing offensive text with natural homoglyph noise. Additionally, the resource you provided, while large, is not focused on homoglyphs for Latin letters and misses potential edge cases such as the runic “ᚹ” character (U+16B9) as a homoglyph for the Latin letter “p” (U+0070). Similar issues associated with other existing normalization tools were identified (see Response A to Reasons to Reject from Reviewer GyN8).
>
> D. The reviewer’s suggestion is well taken. In order to support reproducibility, we will update the paper to include figures and data visualizations such as a histogram of the Cyrillic characters, a histogram of the offensive words, and a mapping of the observed offensive words with homoglyphs to their normalized counterparts. Additionally, we will add information about the fold size and creation process for the 5-fold cross validation used in this study.
>
> E. As we mention in the Related Work section, other datasets containing homoglyphs in offensive language have been created, such as the work done by Kurita et al. (2019). However, our study is the first to produce a dataset composed entirely of natural homoglyph-laden offensive text to our knowledge.
>
> F. The seven models were chosen based on their open-source availability and recent investigations in hate speech detection. We focused on models that were trained (at least in-part) with organically generated data collected from social media platforms—specifically Twitter. We will expand the discussion of our model selection, if the paper is accepted.
>
> G. We agree that comparing monolingual and multilingual models would be interesting and warrants a future study.
>
> H. The complete annotation guidelines provided to the annotators are in Appendix Section A.4.
>
> I. Thank you for the suggested Typos, Grammar, Style, and Presentation Improvements. If accepted, we will adopt your recommendation to relocate Section 3.3 to the Appendix and include more precision for the analysis of the results. As stated in our Ethical Consideration, we will provide the dataset to researchers who request it.

---

### Official Review · Reviewer_GyN8 · 2023-08-04

**Typos Grammar Style And Presentation Improvements:** None
**Soundness:** 2

**Excitement:**

3: Ambivalent: It has merits (e.g., it reports state-of-the-art results, the idea is nice), but there are key weaknesses (e.g., it describes incremental work), and it can significantly benefit from another round of revision. However, I won't object to accepting it if my co-reviewers champion it.

**Missing References:**

None

**Paper Topic And Main Contributions:**

The paper develops a scraping method which can collect hateful/offensive tweets which contain homoglyphs. The authors then annotate a subset of the dataset and show that existing models do not perform well on the new dataset. Upon substituting the Cyrillic characters with their Latin counterparts, the performance improves. The main contributions of the paper are the homoglyphs dataset.

**Questions For The Authors:**

A. As the authors pointed out, a large chunk (88.89%) of the dataset is misogynistic in nature. It would be interesting to see how a misogyny classifier would perform.

B. It is interesting to see the Czech language to be in the majority. Any particular reason for that?

**Reasons To Accept:**

A. The dataset could be a valuable contribution to the fight against hate speech.

B. It's interesting to see that the existing models were not able to perform well in the presence of such homoglyphs.

**Reasons To Reject:**

A. The authors claim that there is currently no library which automates the conversion. This is not true. The Python library unidecode (https://pypi.org/project/Unidecode/) does a pretty good job.

B. The presentation needs to be improved. The authors should provide some sample text for the reviewers to understand the dataset. If possible, a sample should be provided online.

C. The author needs to clearly separate hate speech from offensive speech. Many of the keywords used for the search query might be considered offensive but not hate speech.

**Reproducibility:**

4: Could mostly reproduce the results, but there may be some variation because of sample variance or minor variations in their interpretation of the protocol or method.

**Reviewer Confidence:**

4: Quite sure. I tried to check the important points carefully. It's unlikely, though conceivable, that I missed something that should affect my ratings.

---

> ### Author Rebuttal · Authors · 2023-08-28
>
> We appreciate your thorough analysis of the paper and the opportunity to address your concerns:
>
> Reasons to Reject
>
> A.    While Unidecode performs homoglyph normalization (including Cyrillic to Latin characters), the accuracy varies from homoglyph to homoglyph. For example, the Cyrillic “п” character (U+041F), which is classified as a homoglyph for the Latin “n” character (U+006E) by the Unicode Confusable standard, is normalized by Unidecode to the Latin letter “p” (U+0070). Similar issues were associated with other existing normalization tools (see #3 in response to Reviewer BrA1). For these reasons, automated normalization was not used in the present study.
>
> The poor performance of existing homoglyph normalization tools and our decision not to use one should be explained in greater detail in the paper, and we will add this information, if the paper is accepted. Additionally, we will include an evaluation of the Unidecode library’s normalization capabilities on the annotated sample of our dataset and subsequent performance of the models on this text (normalized by Unidecode).
>
> B.    We agree that providing verbatim tweets would strengthen the paper. If the paper is accepted, we will include tweets which exemplify the annotation categories (e.g. hate speech, references to violence, etc.).
>
> C.    We will add language to the paper to draw a clearer distinction between the capture and application of offensive text and hate speech in the present study. In short, offensive search terms with Cyrillic homoglyphs were used as “seed” to assemble a dataset of hate speech containing a much broader range of homoglyphs. However, the capture of non-hate speech containing offensive words facilitated the evaluation of the models in that it provided negative cases.
>
> Questions
>
> A.    We agree that evaluating the performance of misogyny classifiers would be an interesting addition to the paper. In order to provide an additional zero-shot baseline, we will include results with the model developed by Saha et al. (2018), which had the highest performance on the 'Automatic Misogyny Identification' task at EVALITA-2018 (Paper: https://arxiv.org/abs/1812.06700).
>
> B.    Regarding the misclassification of English text as Czech text when homoglyphs are present—this issue appears to be Twitter specific. Unfortunately, Twitter does not provide information on how it selects language identifiers for tweets, so we cannot provide a definitive answer. It may be related to the global region from which Twitter believes a tweet was posted.

---

### Official Review · Reviewer_KfRS · 2023-08-05

**Soundness:** 4

**Excitement:**

3: Ambivalent: It has merits (e.g., it reports state-of-the-art results, the idea is nice), but there are key weaknesses (e.g., it describes incremental work), and it can significantly benefit from another round of revision. However, I won't object to accepting it if my co-reviewers champion it.

**Paper Topic And Main Contributions:**

In this paper, the authors consider potential hate speech tweets containing homoglyphs. The authors develop a character substitution scraping method to sample offensive tweets with the Homoglyphs (OTH) Dataset. The authors reported that transformer-based hate speech classifiers perform poorly under zero-shot settings. Text normalization improves the performance of these models. The authors also concluded that training the model with labeled data improved its performance substantially.

**Questions For The Authors:**

1)Why did the author choose a sample size of 200 for the annotation?
2) What is the inter-annotator agreement?
3) What is the overall value of Cohen’s Kappa?

**Reasons To Accept:**

- Proposed and benchmark offensive tweet dataset containing homoglyphs.
- Develop a  character substitution scraping method to sample offensive tweets containing homoglyphs



**Reasons To Reject:**

- I do not see any novelty in the proposed classifier models that are used to create the benchmarks,
- The size of labeled data is very small (only 200 tweets).


**Reproducibility:**

5: Could easily reproduce the results.

**Reviewer Confidence:**

4: Quite sure. I tried to check the important points carefully. It's unlikely, though conceivable, that I missed something that should affect my ratings.

---

> ### Author Rebuttal · Authors · 2023-08-28
>
> We greatly appreciate your time and attention, and the opportunity to respond to your questions:
>
> The overall inter-annotator agreement in the annotated sample (n=200) reported in the paper was 97.64%. Inter-annotator agreement by annotation category ranged from 94.5% (hate speech) to 100% (English-language body text). We will add a column to Table 5 in Appendix A.6 to report this data.
>
> The Kappa across all annotation categories was 0.953 (SE = 0.008; 95% CI = 0.937 - 0.969). [Note: This Kappa seems high due to the increase in coding decisions resulting from the cumulative assessment of intercoder reliability, but this number is accurate].  We will add the overall Kappa to the paper.
>
> We agree that the size of the annotated sample was small; however, it was adequate to evaluate the models. Since the submission of the paper to EMNLP, we have annotated an additional 500 tweets, and the results are consistent with those presented in the paper—48.4% of the 500-tweet sample were classified as hate speech (compared with 45.0% of the 200-tweet sample reported in the paper). If the paper is accepted, we can revise Table 5 Appendix in A.6 to include results of all 700 available annotations. Further, we will rerun the evaluation of the models using the 700 annotations.

---

### Meta-Review · Senior_Area_Chairs · 2023-10-04

**Recommendation:** 3

**Metareview:**

Overall Assessment: The paper titled "Offensive Tweets with Homoglyphs: A Benchmark Dataset for Hate Speech Detection" presents a novel dataset and associated experiments for hate speech detection in tweets containing homoglyphs. The reviewers have provided diverse opinions on the paper, which warrant careful consideration.

Scores: Based on the provided reviews, the average Soundness score is approximately 3 (Good, with some minor support needed). The average Excitement score is 3 (Ambivalent, with merits but key weaknesses). These scores indicate a paper with valuable contributions but room for improvement.

---

### Decision · Program_Chairs · 2023-10-07

**Decision:**

Accept-Findings

**Comment:**

Overall Assessment: The paper titled "Offensive Tweets with Homoglyphs: A Benchmark Dataset for Hate Speech Detection" presents a novel dataset and associated experiments for hate speech detection in tweets containing homoglyphs. The reviewers have provided diverse opinions on the paper, which warrant careful consideration.

Scores: Based on the provided reviews, the average Soundness score is approximately 3 (Good, with some minor support needed). The average Excitement score is 3 (Ambivalent, with merits but key weaknesses). These scores indicate a paper with valuable contributions but room for improvement.